# Ag_2_O-Containing Biocidal Interpolyelectrolyte Complexes on Glass Surfaces—Adhesive Properties of the Coatings

**DOI:** 10.3390/polym15244690

**Published:** 2023-12-13

**Authors:** Vladislava A. Pigareva, Oleg S. Paltsev, Valeria I. Marina, Dmitrii A. Lukianov, Andrei V. Moiseenko, Nikita M. Shchelkunov, Andrey A. Fedyanin, Andrey V. Sybachin

**Affiliations:** 1Faculty of Chemistry, Lomonosov Moscow State University, Leninskie Gory, 1-3, 119991 Moscow, Russia; vla_dislava@mail.ru (V.A.P.); ymmo@mail.ru (V.I.M.); dmitrii.a.lukianov@gmail.com (D.A.L.); 2A.N. Nesmeyanov Institute of Organoelement Compounds of Russian Academy of Sciences, Vavilov Street, 28, 119991 Moscow, Russia; 3Skolkovo Institute of Science and Technology, Center for Molecular and Cellular Biology, Bolshoy Boulevard, 30, 121205 Moscow, Russia; 4Faculty of Biology, Lomonosov Moscow State University, Leninskie Gory, 1-5, 119991 Moscow, Russia; postmoiseenko@gmail.com; 5Faculty of Physics, Lomonosov Moscow State University, Leninskie Gory, 1-2, 119991 Moscow, Russia; shchelkunov@nanolab.phys.msu.ru (N.M.S.); fedyanin@nanolab.phys.msu.ru (A.A.F.)

**Keywords:** functional coating, biocide, polycation, polyanion, interfacial complex, coating, nanoparticles, interpolyelectrolyte complex, optical tweezer, adhesion

## Abstract

Biocidal coatings are of great interest to the healthcare system. In this work, the biocidal activity of coatings based on a complex biocide containing polymer and inorganic active antibacterial components was studied. Silver oxide was distributed in a matrix of a positively charged interpolyelectrolyte complex (IPEC) of polydiallyldimethylammonium chloride (PDADMAC) and sodium polystyrene sulfonate (PSS) using ultrasonic dispersion, forming nanoparticles with an average size of 5–6 nm. The formed nanoparticles in the matrix are not subject to agglomeration and changes in morphology during storage. It was found that the inclusion of silver oxide in a positively charged IPEC allows a more than 4-fold increase in the effectiveness of the complex biocide against *E. coli K12* in comparison with the biocidal effect of PDADMAC and IPEC. Polycation, IPEC, and the IPEC/Ag_2_O ternary complex form coatings on the glass surface due to electrostatic adsorption. Adhesive and cohesive forces in the resulting coatings were studied with micron-scale coatings using dynamometry. It was found that the stability of the coating is determined primarily by adhesive interactions. At the macro level, it is not possible to reliably identify the role of IPEC formation in adhesion. On the other hand, use of the optical tweezers method makes it possible to analyze macromolecules at the submicron scale and to evaluate the multiple increase in adhesive forces when forming a coating from IPEC compared to coatings from PDADMAC. Thus, the application of ternary IPEC/Ag_2_O complexes makes it possible to obtain coatings with increased antibacterial action and improved adhesive characteristics.

## 1. Introduction

The use of conventional low-molecular-weight biocides for the treatment of premises with high sanitary standards has a number of significant disadvantages, which can be accompanied by economic costs, as well as potential danger to humans [1,2,3,4,5,6,7,8,9,10,11]. First of all, this is the low durability of the resulting coatings. Generally speaking, solutions of low-molecular-weight biocides, with which walls are treated, have low adhesion to the surface, and therefore such treatment must be carried out more often. In addition, one of the problems is the evolution of bacterial cultures, the emergence of new strains, and the growth of resistance of bacterial microorganisms to disinfectants. [12,13,14]. As a result, there is a possibility that absolute sterilization will not be achieved when surfaces are treated with a low-molecular-weight biocide, which can lead to serious poisoning. In this regard, there is an urgent need to search for new systems that combine properties such as increased biocide and durability.

Polymers are often considered as a material with the ability to form durable coatings [15,16,17]. Biocidal compositions based on water-soluble positively charged interpolyelectrolyte complexes (IPEC) are of potential interest for practical use. Such complexes, firstly, contain a large number of positively charged groups, due to which the polycations in the complex impart biocidal properties. In addition, charged areas can impart good adhesion to hydrophilic surfaces, such as glass [18,19]. In addition, positively charged water-soluble IPECs will contain charge-compensated regions, which will improve adhesion on hydrophobic surfaces. However, it is known that, in general, polycations have lower biocidal activity compared to low-molecular-weight bioactive compounds [20,21,22,23,24,25,26]. In this regard, modifying IPEC by including an additional low-molecular-weight biocidal agent, for example, metal nano particles or metal oxides, is proposed. Among the metals, nano-Ag, nano-Au, nano-Cu, and nano-Se are often considered as an additive [27,28]. Silver and its oxide have proven themselves especially widely.

The bactericidal properties of silver and its derivatives have been repeatedly discussed in the literature and have found application in the creation of wound dressings, plasters, and bandages, as an antibacterial and anticancer agent, and so on [29,30,31,32]. However, most studies where nanoparticles are included in compositions as an antibacterial agent consider the use of an inert polymer matrix [30,33,34]. Therefore, biocidal compositions based on silver (I) oxide nanoparticles and interpolyelectrolyte complexes as a stabilizer are of potential interest for practical use, since such a polymer matrix contains both areas with free charged groups and hydrophobic areas to improve adhesion to hydrophobic surfaces. Moreover, the use of silver (I) oxide in this case is explained by the availability and simplicity of its chemical preparation, as well as the absence of interaction with the polyanion, which makes it possible to achieve a uniform distribution of the resulting nanoparticles in the volume of the material.

Previously, we have already studied the formation and properties of water-soluble IPECs based on pH-independent polydiallyldimethylammonium chloride (PDADMAC), the biocidal properties of which are widely known [35,36], and sodium polystyrene sulfonate (PSS) [18]. The partial neutralization of PDADMAC did not reasonably affect the charge of the resulting IPEC. The obtained IPEC has shown excellent resistance towards dissociation and phase separation in water–salt media formed by various salts. This means that such complexes have been shown to be stable in solutions of high ionic strength, which may allow modification with low-molecular-weight biocides that provide ions in solution. It was shown that a coating formed by the individual polycation could be easily removed by wash-off with water, but the modification of PDADMAC with PSS results in the formation of a coating with good resistance to wash-off. The gradual washing off of coatings from the complexes on a surface will allow the biocidal activity of such a coating to be preserved for a long time—with each cycle of washing off the IPEC layer, the biocidal layer will be regenerated [19].

Assessment of the mechanical properties of coatings is an important task for analyzing the possibility of their practical application. While approaches to studying the adhesion of polymer films to surfaces have long been developed and are actively used [37,38], analysis of the adhesion of macromolecules directly at the interface, as well as the establishment of a correlation between the adhesive properties of individual macromolecules and adhesive properties of polymer films, is still a challenging task.

In this work, IPEC was modified by including a low-molecular-weight biocide—Ag_2_O nanoparticles. The formation of coatings from a ternary IPEC/Ag_2_O composite on a glass surface was studied, and the mechanical and biocidal properties of the resulting coatings were also studied.

## 2. Materials and Methods

### 2.1. Materials

Silver oxide Ag_2_O was purchased from SilverSalt (St. Petersburg, Russia); poly-(diallyldimethylammonium) chloride (PDADMAC) with weight average mass Mw 450,000 and poly-(styrenesulfonate) sodium salt (PSS) with weight average mass Mw 70,000 were purchased from Sigma-Aldrich (St. Louis, MO, USA). All chemicals were used without additional purification. Bidistilled water was used in all experiments.

Borosilicate glass microspheres (BSM) with average diameter 5 µm from Duke Scientific (Palo Alto, CA, USA), and glass slides (GS) and glass cover slips from ThermoFisher (Paolo Alto, CA, USA) were used as test glass surfaces.

IPEC Preparation

The formation of IPEC was performed by the addition of 9.6 mL of 2 wt.% PDADMAC solution in bidistilled water to 86.4 mL of 0.034 wt.% of PSS solution under stirring conditions to provide homogenous distribution of the macromolecules. The homogeneity of the solution was controlled by turbidimetry [39]. As a result, a complex with a molar ratio of monomer units (PSS)/(PDADMAC) = 0.12 was obtained. After this, the complex was concentrated on a vacuum rotary concentrator to 2 wt.% solution.

IPEC/Ag_2_O Preparation

First, 0.4 mg of Ag_2_O powder was added to 2 mL of 2 wt.% of IPEC. The obtained inhomogeneous system was then separated into Ag_2_O precipitate and IPEC solution. Then, the composition was homogenized in a sonic bath with power 500 W for 10 min. The resulted turbid dispersion did not contain visible micro- or macroparticles of silver oxide.

PDADMAC and IPECs Coatings Preparation

Coatings from the polycation and its complexes were obtained using the following method. The pre-cleaned glass substrate was dipped into a 2 wt% polymer solution, kept for 2 min, and then washed with bidistilled water for 2 min to remove excess polymer.

Polymer coatings for microbiological studies were prepared in the following way. A 20 mg/mL aliquot of the polymer solution was applied to the substrate so that the entire surface was covered with the solution, then the substrate was air dried.

BSM covering with PDADMAC and IPECs procedure

First, 2 mg of borosilicate glass microspheres were suspended in 1 mL of methanol and immediately centrifuged. Next, the BSM were resuspended in 1 mL of 1 M KOH and were precipitated again in a centrifuge. After this, the BSM were washed 5 times with distilled water.

Polymer-coated microspheres were prepared as described below. A suspension of 0.5 mg/mL BSM and 0.0675 wt.% PDADMAC was vortexed for 5 min. After that, the polymer-coated BSMs were rinsed 2 times with distilled water to remove residual unbound polymer. Microspheres coated with IPEC were obtained using the method described above.

### 2.2. Methods

The hydrodynamic diameters were measured by dynamic light-scattering (DLS) using Brookhaven ZetaPlus (Brookhaven, Holtsville, NY, USA) equipment with software provided by the manufacturer.

The electrophoretic mobility (EPM) of liposomes and their complexes was controlled by Brookhaven ZetaPlus (Brookhaven, Holtsville, NY, USA) equipment with software provided by the manufacturer.

Transmission electron microscopy (TEM) imaging was performed with JEM-2100 200 kV electron microscope (JEOL Ltd., Tokyo, Japan). Images were acquired with a US1000XP CCD camera (Gatan, Pleasanton, CA, USA) at near-zero defocus. The samples were prepared by applying a drop of an aqueous solution of the substance to a copper grid covered with type B carbon support film (EMCN, Beijing, China), followed by further drying in an air atmosphere. The samples were examined without preliminary contrasting. The series of images was analyzed using ImageJ software (version 1.46r).

Gram-negative *Escherichia coli K12* (*E. coli K12*) strain transformed with “pDualrep2’’ reporter plasmid [20] was used to study the antimicrobial activity of polymer coatings. An overnight culture of cells *E. coli K12*, transformed with “pDualrep2” reporter plasmid, was diluted to 0.05 to 0.1 OD (590 nm) units with fresh TSB (Trypticasein Soy Broth) medium supplied with 50 μg/mL ampicillin and plated on TSB agar medium (50 μg/mL ampicillin). After plate drying, glass covered with PDADMAC, IPEC, or IPEC/Ag_2_O was placed on top of the agar. Alternatively, instead of placing glasses covered with complexes, 2 µL of PDADMAC, IPEC, or IPEC/Ag_2_O at a concentration of 20 mg/mL was dropped on the plates on the substrate of bacteria. Then, plates were incubated overnight at 37 °C. The agar plate was scanned using the ChemiDoc Imaging System (Bio-Rad, Boulder, CO, USA) in channels “Cy3-blot” (553/574 nm) and “Cy5-blot” (588/633 nm).

The estimation of minimal inhibitory concentration (MIC) values was carried out for the Gram-negative Escherichia coli K12 strain in TSB medium. The MICs in TSB medium were determined using a serial dilution assay [40]. The cell concentration was adjusted to approximately 5 × 10^5^ cells/mL. PDADMAC, IPEC, and IPEC/Ag_2_O solutions with an initial concentration of 20 mg/mL were used as the test compound. Two-fold serial dilution of the polymers solutions was performed in a 96-well microplate. The microplates were covered and incubated at 37 °C with shaking. The OD600 of each well was measured, and the MIC was assigned as the lowest concentration of the tested compound that resulted in no growth after 16–20 h. Bacterial cell growth was measured at 590 nm using a microplate reader (VICTOR X5 Light Plate Reader, PerkinElmer, Waltham, MA, USA).

The surface of a clean glass substrate, as well as polymer coatings, was characterized by determining the contact angle using TRACKER^TM^ Standard Drop Tensiometer (Teclis Scientific, Lyon, France).

The study of surface roughness of coatings was performed using scanning probe microscope Nanoscope IIIa (Veeco, Santa Barbara, CA, USA) with silicon cantilevers with resonance frequencies 140–150 KHz (TipsNano, Zelenograd, Russia) in tapping mode. The preparation of samples is described elsewhere [19]. The analysis of root mean square roughness (Sq) was performed with the Gwyddion software package (version 2.62).

Mechanical tests of the coatings

The dynamometric measurements were performed according to the procedure described elsewhere [41]. The 2 wt.% IPEC solution was applied on the freshly cleaned GS so that the entire surface was covered with the solution. Two minutes later, the IPEC solution was removed, and the resulting GS with modified surface was washed with bidistilled water. Another GS was placed upon the resulting coating so that the “sandwich” structure GS-IPEC-GS was formed. After 24 h of drying, the adhesive properties of the IPEC were studied by dynamometry using a tensile testing machine by Metrotest (Moscow, Russia). The values of the stress required to separate the “sandwich” structure were collected.

Optical tweezers measurements

The laser tweezers setup was described elsewhere [42]. The method of optical tweezers was used to study the adhesion properties of the polymer complex at the microscopic level. The microscale process visualization in the optical trap was carried out using a high-aperture objective lens and a SMOP camera. The position of the traps was controlled by a movable lens and a servo-driven coordinate stage. The interaction of PDADMAC or IPEC with the glass surface was studied by the determination of forces needed to break the polymer-to-surface interfacial complex.

Experiments were carried out in the following way. To make a chamber with a studied suspension of polycationic brushes with BSMs, two coverslips were taken. The coverslips were treated with surface-active substance, then carefully washed with water and treated with hydrogen peroxide. After the coverslips dried, the surface of one of them was covered with a layer of albumin to prevent the adhesion of the particles in suspension to the surface of the coverslip. Furthermore, 45 μL of aqueous suspension of the studied sample was placed in the gap between the two coverslips. The prepared chamber was placed on an objective table. Two optical traps were formed within the sample. Complex of PDADMAC or IPEC with the BSM was captured by traps. Then, the distance between the centers of the traps was increased, creating an external force to break this complex. The maximum optical force value of the trap corresponded to the power at which the break took place.

This force was detected using the force calibration based on the drag force method [43]. Laser tweezers allow one not only to capture various micro-objects and manipulate their position in space, but also to carry out quantitative measurements of the forces that act on these micro-objects. For this purpose, the calibration of the experimental setup of optical tweezers was made by the following means of viscous friction. As is known, a viscous friction force acting on a spherical particle with radius *a* is proportional to the particle’s speed *u* and determined by the Stokes law by the relation:*F_vis_* = −6*πηau*(1)
where the parameter *η* is the dynamic viscosity of the medium. In order to calibrate the force of the optical tweezers with this force *F_vis_*, a micro-object with the determined dimensions was taken and placed in a medium whose dynamic viscosity was also known. Then, an optical trap was formed at the location of this micro-object. As soon as the microparticle is trapped, the latter begins to move at a constant speed in the same plane in which the capture took place. The last action was repeated until there was such a value for the speed of the trap at which, at high speeds, micro-objects would fly out of the waist of the laser trap. For this speed value, the following condition takes place:*F_vis_* = −6*πηau* = *F_trapmax_*(2)

In other words, the force of viscous friction will be equal to the maximum force of optical capture of the micro-object with laser tweezers for the selected radiation power in the waist.

An attempt to break the complex is made at the different maximum optical trapping forces (*F*_trap*max*_). By incrementally increasing the laser power, a force that allows the complex to be separated is achieved. Since the viscous friction force is known, using the calibration described above, the binding force of the complex can be determined. In the statistical analyses, the average results of at least five experiments are presented as mean values.

## 3. Results and Discussion

### 3.1. IPEC/Ag_2_O Complexes Characterization

Interaction of nanoparticles with polyelectrolytes could induce the formation of gel structures where nanoparticles act as crosslinkers of the network or nanoparticles could be distributed within macromolecules coils [42,44,45]. That is why, at the first step, we have investigated the distribution of the Ag_2_O particles in the dispersion of IPEC/Ag_2_O using DLS and TEM measurements. The mean diameter of the complexes determined by DLS was found to be 260 nm. This value corresponded to the size of the bare IPEC without Ag_2_O inclusions. Therefore, no aggregation of the IPECs induced by the formation of bonds with Ag_2_O particles was detected and ternary complexes of PDADMAC/PSS/Ag_2_O could be considered as individual IPECs with incorporated nanoparticles of Ag_2_O.

In order to analyze the particle size of silver oxide particles in the complex and their intra-complex distribution, TEM images were obtained. A typical microphotograph is presented in Figure 1. Individual contrast Ag_2_O nanoparticles with mean diameter 5.5 +/*−* 0.5 nm could be detected on the image. Amorphous chains of the polyelectrolytes in the IPEC could not be detected without additional contrast.

### 3.2. Biocidal Properties of IPEC/Ag_2_O

The antibacterial properties of IPEC/Ag_2_O were investigated on agar plate with TSB with a substrate of Gram-negative *E. coli K12* strain. To begin with, drop tests with PDADMAC, IPEC, and IPEC/Ag_2_O were carried out, and a clean glass substrate was examined. The results of the experiment can be seen in the Appendix A). It was found that the polymer solutions showed a zone of inhibition of bacterial growth. For a clean glass substrate, there was no inhibition zone.

Then, cover slips with an identical shape and area coated with PDADMAC, IPEC, and IPEC/Ag_2_O were deposited on the test surface in a Petri dish, so that the polymeric layer was in direct contact with the bacteria. In Figure 2, the areas of inhibition zones for PDADMAC (a), IPEC (b), and IPEC/Ag_2_O (c) are presented. All samples have demonstrated antibacterial activity—dark areas reflect death of *E. coli*. The slightly larger area for the PDADMAC sample could be attributed to the more effective migration of polycationic macromolecules from the coating in agar in comparison to migration from IPEC and IPEC/Ag_2_O coatings. Contrast images of glass boundaries and inhibition zones are given in the Appendix A. The differences in the diffusion of polymer chains are due to hydrophobization of the IPECs by PSS molecules, and thus the coatings on the glass are more stable with regard to the diffusion of PDADMAC and PSS molecules from the coating layer to agar. This result is in good agreement with data on the resistance of the polymer coatings from pure polycation and IPECs towards wash-off with water [18].

In addition, MICs were estimated using Gram-negative *E. coli K-12* in TSB media (Table 1). While PDADMAC and IPEC have demonstrated similar biocidal activity, the incorporation of Ag_2_O nanoparticles into IPEC resulted in a more than 4 times decrease of MIC, from more than 0.4 mg/mL to 0.1 mg/mL, reflecting an increase of antibacterial activity of the composition. It seems that modification of 12 mol.% of cationic units with PSS does not dramatically change the antibacterial activity of PDADMAC. According to the structure of the interpolyelectrolyte complexes discussed in the literature, the IPECs formed at a specific molar ratio of polycations and polyanions do not form the complex where all charged units of the polyelectrolyte that is added in deficiency to opposite charged polyelectrolyte form salt bonds. In fact, due to the non-complementary structure of polymers, the number of salt bonds is smaller than could be directly calculated from the molar ratio of anionic to cationic units [46]. Thus, the real number of unbound PDADMAC units that ensure biocidal action is more than 90%.

### 3.3. Shelf-Life of IPEC/Ag_2_O Complexes

Storage of the dispersion over several days resulted in precipitation of IPEC/Ag_2_O complexes. Therefore, questions about colloid stability, possible sediment aging, or evolution of complexes during storage arise. First, the colloid stability of the freshly prepared dispersion was studied over 12 h by means of DLS. No change in size of the complexes was detected. Then, the sample was analyzed after 14 days of shelf life. After two weeks of incubation at room temperature, the formation of a sediment of IPEC/Ag_2_O was observed. Vigorous shaking of this sediment resulted in redistribution of the IPEC/Ag_2_O complexes and formation of a uniform turbid dispersion similar to the freshly prepared one. The redispersed IPEC/Ag_2_O complexes were investigated using DLS and TEM. The mean diameter of the complexes in the water dispersion was found to remain 260 nm. A TEM microphotograph of the complex is presented on Figure 3. The average size of the individual Ag_2_O nanoparticles was found to be 6.0 +/*−* 5 nm. Therefore, no evolution of nanoparticles or complexes took place during storage.

Therefore, we can conclude that the sedimentation of IPEC/Ag_2_O complexes is due not to the aggregation of particles, but to the action of gravitational forces. In addition, distribution of the nanoparticles in the IPEC matrix remains uniform, and no aggregation or change of the morphology of nanoparticles takes place.

### 3.4. Formation of Polymer Coatings on Glass Surface

The formation of adsorption layers of PDADMAC, IPEC, and IPEC/Ag_2_O on a glass surface was studied by measuring the EPM of the suspension of BSM with added polymer solutions.

The titration curves are presented in Figure 4. The addition of PDADMAC to the suspension of BSM resulted in an increase in the EPM values from −3 (µm/s)/(V/cm) to +1.3 (µm/s)/(V/cm) due to the formation of electrostatic complexes and charge compensation (Figure 4 curve 1). The point with an electroneutral EPM at PDADMAC concentration 6 × 10^−6^ base-mol/L (Figure 4 point A) corresponds to complete surface charge neutralization. Reaching the positive charge plateau of the titration curve reflects the termination of the PDADMAC adsorption due to electrostatic repulsion between polycation and recharged BSMs. Such behavior of the system is typical for the adsorption of polycations on anionic colloid particles [47,48,49,50,51]. Substitution of PDADMAC with IPEC resulted in a shift of the titration curve to higher concentrations of cationic units of PDADMAC, which is required to neutralize the surface charge of BSM—7.5 × 10^−6^ base-mol/L of PDADMAC in IPEC (Figure 4 point B). This is due to fact that PSS neutralizes 12% of the charged units of PDADMAC, which are responsible for binding with the negatively charged BSM surface and its recharging. In addition, it is important to stress that the 12% shift of the neutralization point for IPEC in comparison to pure PDADMAC reflects that all PSS molecules remained in the adsorbed IPEC and were not eliminated from the complex by competitive interaction with anionic surface [52,53]. The incorporation of Ag_2_O nanoparticles into the complex did not lead to a change in the behavior of adsorption of the ternary IPEC/Ag_2_O complex on BSM compared to the binary IPEC—curves 2 and 3 in Figure 4 coincide.

Thus, to study the adhesive properties of coatings further, we will consider only PDADMAC and binary IPECs.

### 3.5. Mechanical Properties of PDADMAC and IPEC Coatings

The adhesion of the PDADMAC and IPEC coatings on the glass surface were studied with dynamometry. Cohesive forces of the coatings were studied by estimation of the peak stress values required to break the polymer “bridges” between two glass slides in the lateral direction, while adhesive forces of the coatings were estimated in the perpendicular direction of the application of the stress. The values of the peak stress are presented in the Table 2.

Lateral stress is caused by cohesive forces, while perpendicular stress is mainly caused by adhesive forces. It should be noted that adhesive forces with lower values of stress should be considered as determining the properties of coatings. Thus, it can be assumed that coatings formed from a polycation and an interpolyelectrolyte complex have similar adhesive and cohesive properties. The explanation could be as follows. IPEC has an amphiphilic structure, where charged units provide hydrophilic properties and areas of bound PSS and PDADMAC units are responsible for hydrophobic areas. Adsorption from the water media on the glass surface results in the formation of an interfacial complex with cationic units that are in molar excess, expanded to the water media, forming loops and tails. IPEC has a dynamic internal structure, where macromolecules are capable of moving along the opposite charged chains. Hydrophobic fragments of the IPEC are expected to be situated closer to the surface to minimize the surface tension. As a result, the outer layer of the IPEC represents itself almost as individual PDADMAC. Hence, the peak stress values obtained in dynamometry experiments could be attributed to breakage of the contacts between the coating and the upper glass slide that interact with the top layer of the adsorbed PDADMAC and IPEC. However, these results demonstrate that formation of the IPEC (i.e., hydrophobization of the PDADMAC) does not result in loss of the adhesive properties of the formed coating.

In order to support the assumption above, the contact angles of water on the coatings were studied. The presence of a significant share of inorganic particles of Ag_2_O should cause the distinguishable change of the water contact angle on the surface of the polymer films [54]. The freshly cleaned glass had a contact angle of 30°, and formation of the coating from PDADMAC resulted in slight hydrophilization of the surface (the contact angle was found to be 20°). The formation of coatings from either IPEC or IPEC/Ag_2_O did not result in a change of contact angle in comparison to the PDADMAC coating. Because of the fact that surface relief could affect the contact angle, the root mean square roughness (Sq) values were measured (see Appendix A Appendix A). Sq values for coatings from polycation and IPEC were found to be almost identical—2.5 nm and 2.3 nm, respectively. Therefore, one may assume that the top layers of the coatings have similar chemistry. Hence, PSS macromolecules do not present on the surface of the coatings, otherwise hydrophobization of the layer and an increase of the contact angle should be observed.

Therefore, in order to evaluate the real adhesion properties of the interpolyelectrolyte complexes on a glass surface, it is necessary to use a method capable of recording the properties of individual macromolecules.

### 3.6. Mechanical Properties of PDADMAC and IPEC Coatings on Submicron-Scale

The use of optical tweezers makes it possible to estimate the force required to detach the interpolyelectrolyte complex from the glass microsphere surface. For this purpose, some borosilicate microspheres coated with the polymer and the interpolyelectrolyte complex were obtained. The coated microspheres were brought into contact with unmodified microspheres using optical tweezers, resulting in a complex of two microspheres linked through the polymer/interpolyelectrolyte complex. Next, a force was applied to the resulting complex to separate the two microspheres (*F_det_*), in other words, tearing the polymer off the surface of one of the microspheres. Thus, the resulting value characterizes the adhesive force of the polycation/interpolyelectrolyte complex. The experimental design is shown in Figure 5.

The optical tweezers study revealed that the force required for the detachment of two microspheres connected via a PDADMAC “bridge” is (37 ± 3) pN. This force value could be attributed to the characteristic of the adhesive force of the individual polycation macromolecule on the glass surface. Modification of the PDADMAC with PSS forming IPEC resulted in an increase of the force required to break the connection between the two BSMs connected via IPEC “bridge”. This value overcomes 118 pN—the limiting value of the experimental setup. It is generally accepted that adsorption of the polycation at the glass surface is irreversible due to high electrostatic binding constant [55,56,57]. With the help of the optical tweezer, we have demonstrated that the interaction force could be quantified and that the electrostatic binding is not the only factor that ensures strong adhesion of the polyelectrolyte. Decrease of the number of cationic units that are capable of forming bonds with the surface with the modification of PDADMAC with 12 mol.% of PSS was expected to result in a decrease of the adhesive forces in the case of dominant electrostatic interaction. However, the results of the experiment demonstrate that hydrophobization of the PDADMAC resulted in an increase of the adhesion of the complex on a hydrophilic surface.

## 4. Conclusions

Since cationic polymers have already proven themselves to be effective antibacterial chemicals [36,58,59,60], for which, unlike classical antibiotics, microorganisms do not develop resistance, the key problem for systems based on them remains the production of coatings with good adhesion to the surface being treated. Traditionally, in practice, the surfaces being treated are quite heterogeneous and may contain both hydrophilic and hydrophobic areas, which can arise as a result of contamination or damage to the original surfaces. Therefore, among polymer biocides, amphiphilic macromolecules are of the greatest interest, since they have an affinity for surfaces with different levels of hydrophobicity. Interpolyelectrolyte complexes can be considered as a class of amphiphilic copolymers, where free charged groups form hydrophilic regions, and regions of salt bonds between the polycation and polyanion form hydrophobic blocks. The production of IPEC has an advantage over the production of traditional amphiphilic copolymers by directed synthesis, since it has greater variability of components and compositions, as well as the possibility of using widely available commercial polymers. In this work, we focused on the study of a water-soluble interpolyelectrolyte complex based on PDADMAC and PSS, for which the ability to form effective coatings on both hydrophilic glass surfaces and hydrophobic polycarbonate surfaces was previously demonstrated [18]. Adsorption of the IPEC on the glass surface is governed by electrostatic forces as well as adsorption of PDADMAC. The formed coatings contain free PDADMAC cationic chain fragments expanded to be external to the treated surface layer, ensuring biocidal activity. Investigation of adhesive and cohesive forces in the resulting coating revealed that the stability of the coating is determined primarily by adhesive interactions. At the macro level, it is not possible to reliably identify the role of IPEC formation on adhesion. On the other hand, the use of the optical tweezers method makes it possible to analyze macromolecules at the submicron scale and to evaluate the multiple increase in adhesive forces when forming a coating from IPEC compared to coatings from PDADMAC. With respect to the biocidal function of the coatings, we may speculate that even if the thick film structure of polymer/IPEC coating will be violated either by mechanical damage or due to wash-off with water, the first layer of the adsorbed macromolecules will remain and will retain the protective antibacterial properties of the coating.

The introduction of 1 wt.% silver oxide nanoparticles into IPEC by ultrasonic dispersion of Ag_2_O powder in a solution of the IPEC resulted in the formation of a stable system of IPEC/Ag_2_O with uniformly distributed 5–6 nm in diameter nanoparticles in the polymer matrix. Formation of such a ternary complex made it possible to increase the antibacterial properties of the initial IPEC by more than 4 times against *E. coli K12*. At the same time, the introduced Ag_2_O nanoparticles did not affect the nature of adsorption of the complex on glass microspheres.

Thus, the formation of ternary IPEC/Ag_2_O complexes makes it possible to obtain coatings with increased antibacterial action and improved adhesive characteristics.

## Figures and Tables

**Figure 1 polymers-15-04690-f001:**
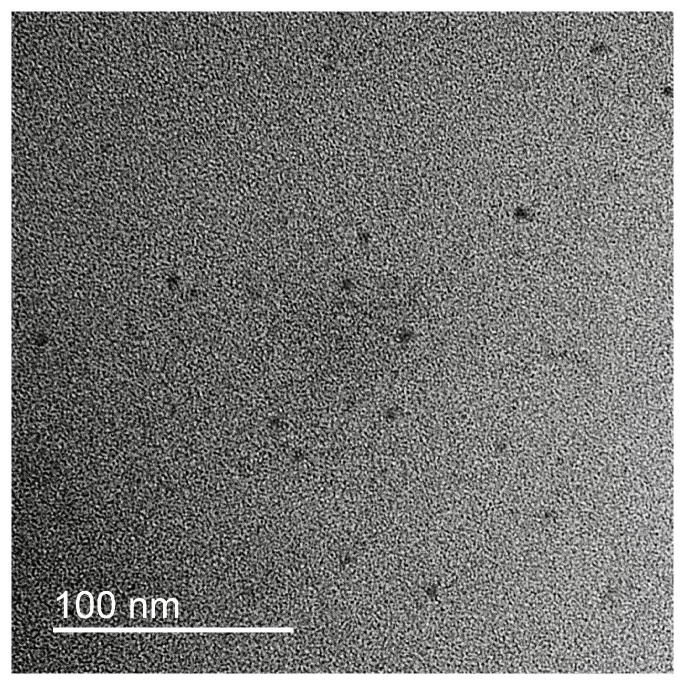
TEM image of freshly prepared IPEC/Ag_2_O.

**Figure 2 polymers-15-04690-f002:**
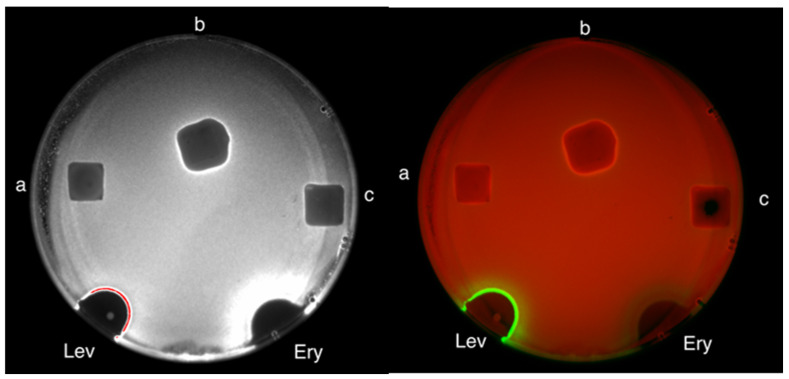
Screening for bacterial growth inhibition of the lawn of *E. coli K12* strains with immersed glass substrates coated with PDADMAC (a), IPEC (b), IPEC/Ag2O (c), spots of erythromycin (Ery) and levofloxacin (Lev) are demonstrated as controls. The plate scanned in the «cy5-blot» channel (left image) for the detection of red fluorescence (Katushka2S). The plate scanned in the «cy3-blot» channel and in the «cy5-blot» channel (right image) for the detection of green fluorescent protein and Katushka2S detection respectively.

**Figure 3 polymers-15-04690-f003:**
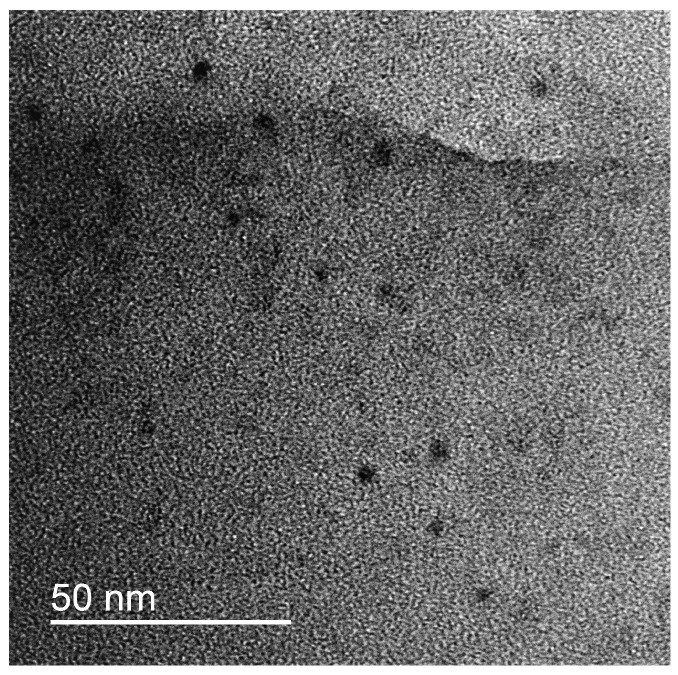
TEM image of IPEC/Ag_2_O after two weeks of storage.

**Figure 4 polymers-15-04690-f004:**
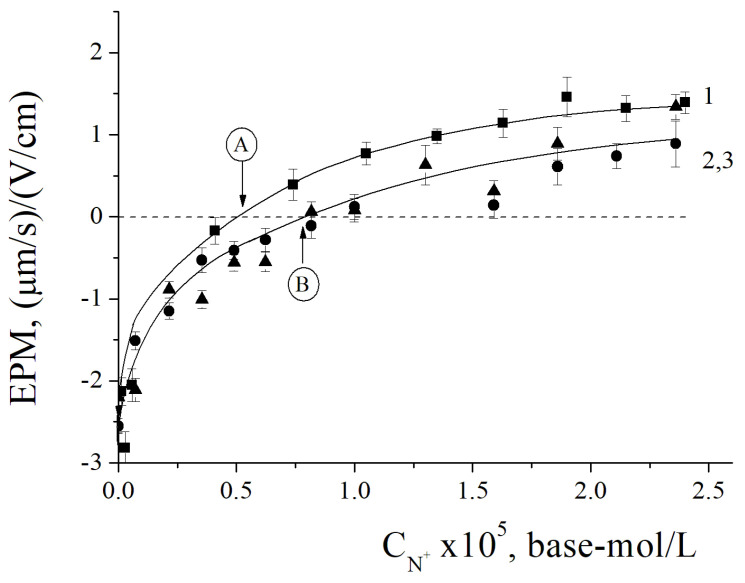
The dependence of EPM of BSM upon the concentration of cationic units in added PDADMAC (1, square), IPEC (2, circle), and IPEC/Ag_2_O (3, triangle). Neutralization point for PDADMAC/BSM suspension (A); neutralization point for IPEC/BSM and IPEC/Ag_2_O/BSM suspensions (B).

**Figure 5 polymers-15-04690-f005:**
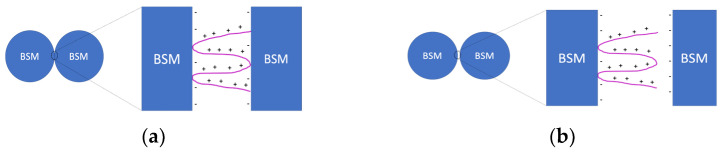
Scheme of the optical tweezers’ measurements. (anionic) BSM-(cationic) polymer- (anionic) BSM complexes separation via applied force by optical trap: (**a**) the force applied below *F_det_*_._; (**b**) the force applied above *F_det_*.

**Table 1 polymers-15-04690-t001:** MIC values for the polymer samples.

Sample	MIC, mg/mL
PDADMAC	>0.4
IPEC	>0.4
IPEC/Ag_2_O	0.1

**Table 2 polymers-15-04690-t002:** Mean peak stress values for the PDADMAC and IPEC coatings on the glass surface.

Coating	Lateral Peak Stress, Pa	Perpendicular Peak Stress, Pa
PDADMAC	99,000 ± 10,000	52,000 ± 13,000
IPEC	139,000 ± 43,000	56,000 ± 8000

## Data Availability

Data available on request.

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
