# Peer review of "Ag2O-Containing Biocidal Interpolyelectrolyte Complexes on Glass Surfaces—Adhesive Properties of the Coatings"

_polymers, 2023, doi:10.3390/polym15244690_

Round 1
Reviewer 1 Report
Comments and Suggestions for Authors
1. The experiment is not fully described on lines 150-155: What is the quantity and density of a microbial suspension of E.coli? It is not TSB, it is agar medium.
2. What is the size of the agar plate and the cover glass covered with polymers - In Figure 2 they seem different and the diffusion zones could not be compared.
3. Delete the "incubated" at the end of line 285.
4. What is presented in Figure 3 - fresh or old redispersed polymer?
5. On line 321 the number of figure is wrong - it was written 5, but it is written 4.
6. On line 327 the title "Mechanical properties of IPEC/Ag2O coatings" should be changed as they investigate only PDADMAC and IPEC coatings.
7. on line 375 the number of the figure is 5, not 4.
8. The article
Hyperbranched Kaustamin as an antibacterial for surface treatment is not accessible
Comments on the Quality of English Language
I don't think I'm good enough to comment on spelling, the text is readable and understandable
Author Response
Please, find below our answers.
- The experiment is not fully described on lines 150-155: What is the quantity and density of a microbial suspension of E.coli? It is not TSB, it is agar medium.
- The experiment is not fully described on lines 150-155: What is the quantity and density of a microbial suspension of E.coli? It is not TSB, it is agar medium.
Answer. We appreciate the reviewer's careful reading of our article. The bacteria lawn plates were prepared using TSB (Trypticasein Soy Broth) with agar instead of the classic LB (Luria broth). To clarify the description we have expanded this section.
2. What is the size of the agar plate and the cover glass covered with polymers - In Figure 2 they seem different and the diffusion zones could not be compared.
Answer. We have used Petri dishes with the following parameters: 92 mm x16 mm. The glass samples’ dimensions were 9 mmx9 mm. The Figure 2 shows a slightly larger zone from PDADMAC glass, which can be attributed to the slightly better migration of PDADMAC in the agar medium. The new image with highlighted areas of inhibition was used to substitute Figure 2. The original photo was added to supplementary materials (see Figure S2).
3. Delete the "incubated" at the end of line 285.
Answer. Correction was made.
4. What is presented in Figure 3 - fresh or old redispersed polymer?
Answer. The figure shows resuspended nanoparticles after two weeks of storage. The correction was made.
5. On line 321 the number of figure is wrong - it was written 5, but it is written 4.
Answer. Image number was corrected
6. On line 327 the title "Mechanical properties of IPEC/Ag2O coatings" should be changed as they investigate only PDADMAC and IPEC coatings.
Answer. The correction was made
7. on line 375 the number of the figure is 5, not 4.
Answer. The Figure number was corrected
8. The article
Hyperbranched Kaustamin as an antibacterial for surface treatment is not accessible
Answer. This could be due to the temporary troubles on journal site. We add the pdf-file of the manuscript and its supplementary to the answer to Reviewer. Also, we expand the number of references that allow one to conclude that MIC values for polymer biocides are few magnitudes higher than for conventional biocides of non-polymeric nature (Ref. 20-26).
We thank the reviewer for important remarks and questions that helped us to improve the manuscript.

Reviewer 2 Report
Comments and Suggestions for Authors
The manuscript entitled “Ag2O-containing biocidal interpolyelectrolyte complexes on glass surfaces – adhesive properties of the coatings” describes an interesting modification route of polyelecrolyte in order to increase biocidal activity. Although the work is original, I have several comments, remarks and suggestions:
1. The novelty of the work compared to the existent publications was not highlighted.
2. The scientific aim of the work was not clearly defined
3. Figure 3: The caption should be corrected.
4. Page 9, second paragraph: the statement “The formation of coatings from either IPEC or IPEC/Ag2O did not resulted in change of contact angle in compare to PDADMAC coating reflecting that the top layers of the coatings have similar chemistry. Hence, PSS macromolecules do not present on the surface of the coatings otherwise the hydrophobization of the layer and the increase of the contact angle should be observed.” should be confirmed experimentally, because differences in contact angle values may also result from other factors, e.g. surface roughness.
5. The two figures are designated as “Figure 4”: on page 8 and on page 9.
6. Figure 4 on Page 8: the presented data (dependence of EPM of BSM upon the concentration of investigated samples) are not clear.
Author Response
- The novelty of the work compared to the existent publications was not highlighted.
- Thescientific aim of the work was not clearly defined
Answer to questions 1&2.
The general aim of this research was to estimate the adhesive properties of the coatings prepared from polycation and water-soluble interpolyelectrolyte complexes. The secondary task of the research was to evaluate the correlation in adhesive properties of the film on the substrate and the adhesive behavior of the first layer of adsorbed macromolecules. While the methods of analysis of cohesive and adhesive properties of polymer films on the substrate are well defined, the analysis of interaction of individual macromolecule or IPEC with the surface is still challenging. It is generally accepted that adsorption of the polycation at the glass surface is irreversible due to high electrostatic binding constant . With the help of optical tweezer we have demonstrated that the interaction force could be quantified and that the electrostatic binding is not the only factor that ensures strong adhesion of the polyelectrolyte. Decrease of the number of cationic units that are capable to form bonds with the surface with the modification of PDADMAC with 12 mol% of PSS was expected to result in decrease of the adhesive forces. However, the results of experiment demonstrate that hydrophobization of the PDADMAC resulted in increase of the adhesion of the complex on hydrophilic surface. At the same time the measurements of the mechanical properties of the films on the glass surface have demonstrated that the modification of PDADMAC with PSS does not significantly change the adhesive/cohesive characteristics of the coating. The cohesive forces were found to be the dominant ones and the impact of the hydrophobization of PDADMAC on cohesive interactions was demonstrated to be negligible. With respect to biocidal function of the coatings we may speculate that even if the thick film structure of polymer/IPEC coating will be violated either by mechanical damage or due to wash-off with water the first layer of the adsorbed macromolecules will remain and will retain protective antibacterial properties of the coating.
The introduction, discussion and conclusion sections were expanded and the corresponding references were added.
3. Figure 3:The caption should be corrected.
Answer. Correction was made.
4. Page 9, second paragraph: the statement “The formation of coatings from either IPEC or IPEC/Ag2O did not resulted in change of contact angle in compare to PDADMAC coating reflecting that the top layers ofthe coatings have similar chemistry. Hence, PSS macromolecules do not present on the surface of the coatings otherwise the hydrophobization of the layer and the increase of the contact angle should be observed.” should be confirmed experimentally, because differences in contact angle values may also result from other factors, e.g. surface roughness.
Answer. Indeed, the contact angle by itself can not be used as the only factor determining the hydophobicity of the surface. We have measured the surface relief of the coatings from PDADMAC and IPEC using AFM. The surface roughness was analyzed using Gwyddion software package. The root mean square roughness (Sq) values were found to be almost identical - 2.5 nm and 2.3 nm for PDADMAC and IPEC coatings respectively. Also, we substitute the statement of the similar chemistry of the surface on the assumption on the similarity in hydrophobicity/hydrophilicity of the top layer of the coatings. The text was updated with AFM data.
5. The two figures are designated as “Figure 4”: on page 8 and on page 9.
Answer. The captures were corrected.
6. Figure 4 on Page 8:the presented data (dependence of EPM of BSM upon the concentration of investigated samples) are not clear
Answer.
We have changed the axis legend of the Figure 4 to show that we compare the similar concentrations of PDADMAC units in pure polycation and IPECs.
The data on the Figure 4 is presented to support two general observations. First one is the fact that modification of the interaction of IPEC with BSM does not result in release of the PSS from the complex via concurrent reaction between negatively charged surface and polyanion. The formation of complex BSM/PDADMAC/PSS is supported with the fact that the point of electroneutrality for the PDADMAC (point A on the Figure 4) is reached at smaller concentrations than for IPEC (point B on the Figure 4). In case of substitution of PSS/PDADMAC bonds with BSM/PDADMAC bonds the curves 1 and 2 should coincide. The second point that we want to demonstrate is that addition of Ag2O nanoparticles to IPEC does not dramatically change the adsorption of complexes on BSMs and the curves 2 and 3 coincide.
We thank the Reviewer for helpful questions and remarks.
Reviewer 3 Report
Comments and Suggestions for Authors
Present work describes the biocidal activity of coatings based on a complex biocide containing polymer (matrix of a positively charged interpolyelectrolyte complex (IPEC) of polydiallyldimethylammonium chloride (PDADMAC) and sodium polystyrene sulfonate (PSS) and inorganic active antibacterial component Silver oxide. Silver oxide was distributed in a matrix of polymer by ultrasonic dispersion, forming nanoparticles with an average size of 5-6 nm. It was found that the inclusion of silver oxide in a positively charged IPEC allows a 4-fold increase in the effectiveness of the complex biocide against E. coli K12 in comparison with the biocidal effect of PDADMAC and IPEC.
IPEC/Ag2O material was characterized by dynamic light-scattering and transmission electron microscopy. Mechanical tests of the coatings, optical tweezers measurements were carried out.
Screening for E. coli K12 growth inhibition was performed. It was found what incorporation of Ag2O nanoparticles into IPEC resulted in 4-times decrease of MIC from 0.4 mg/mL to 0.1 mg/mL compares to PDADMAC and IPEC.
Overall, the material is presented clearly. All synthesized substances are correctly identified. The manuscript is written clearly and understandably without frills. All conclusions supported by the results.
There is one note to the article. The most important parameter of antibacterial activity of biocide is the minimum bactericidal concentration (MBC). Why the authors didn't define this parameter?
Author Response
There is one note to the article. The most important parameter of antibacterial activity of biocide is the minimum bactericidal concentration (MBC). Why the authors didn't define this parameter?
Answer.
The general aim of our research was to estimate mechanical properties of the coatings. Thus, the microbiology experiments were prepared to visualize the antibacterial properties of the coatings. Indeed, the complete analysis of the biocidal properties should include both MIC and MBC values. But in these experiments we made the preliminary measurements to understand whether complexation of PSS will drastically reduce the antibacterial activity or could we increase the antibacterial activity of polymers by incorporating Ag2O nanoparticles. We focused on setting up an experiment to determine the antibacterial properties of polymers, and we chose MIC as an effective method to achieve this goal. Our further investigations of the biocidal coatings will include the complete set of the experiments. Thank you for the important remark.
Reviewer 4 Report
Comments and Suggestions for Authors
The present paper present valuable information but before publications the authors should make some improvements. In my opinion the authors should add in the manuscript the results of the tape-pull test results. Also, on the obtained coatings the authors should perform water contact angle measurements and swelling studies and their results must be added in the manuscript (please see the next paper: https://doi.org/10.3390/coatings13020472). Moreover, the results of the SEM studies should be added in the manuscript.
Author Response
The present paper present valuable information but before publications the authors should make some improvements.
In my opinion the authors should add in the manuscript the results of the tape-pull test results.
Answer: The experiments with tape-pull test were performed but the instrument limitations did not allow us to obtain numerically significant results. Thus, for the system under study, the selection of conditions for the tape-pull experiment is a separate serious task that is beyond the scope of this study.
Also, on the obtained coatings the authors should perform water contact angle measurements and swelling studies and their results must be added in the manuscript (please see the next paper: https://doi.org/10.3390/coatings13020472).
Answer: the contact angles were measured. Please see the section 3.5. The swelling (water absorption by the films) of coatings from PDADMAC was measured in our previous papers (see Coatings 2023, 13(6), 1076; https://doi.org/10.3390/coatings13061076 [ref.41] and Physchem 2023, 3(1), 147-155; https://doi.org/10.3390/physchem3010011). Now we work on complex study of the influence of humidity on mechanics of films, glass temperatures etc. of films composed of IPECs of different composition. Our preliminary results demonstrate no significant difference in water absorbance by films formed by PDADMAC and its IPEC that is under investigation in current manuscript.
The recommended reference was added into Results and Discussion section in subsection 3.5.
Moreover, the results of the SEM studies should be added in the manuscript.
Answer: the SEM images of films from PDADMAC and IPECs were published in previous paper Polymers 2022, 14, 1247 https://doi.org/10.3390/polym14061247 ref.18. The surface of these coatings is smooth and there is no need in these data in current manuscript. At the same time TEM imaging allows us to detect morphology and the size of the Ag2O nanoparticles, that is why we prefer to present here only TEM images.
Thank you for the questions and suggestions.
Round 2
Reviewer 1 Report
Comments and Suggestions for Authors
I am satisfied with the improvements made by the authors and supplementary materials.
Author Response
Thank you for the cooperation.
Reviewer 2 Report
Comments and Suggestions for Authors
The manuscript entitled “Ag2O-containing biocidal interpolyelectrolyte complexes on glass surfaces – adhesive properties of the coatings” has been corrected according my suggestions and all questions have been explained.
Author Response
Thank you for the cooperation.
Reviewer 4 Report
Comments and Suggestions for Authors
Accept in present form